# Genome-wide analysis of fitness determinants of *Staphylococcus aureus* during growth in milk

**Marita Torrissen Mårli[1], Anne Olsdatter Ohren Nordraak[1], Vincent de Bakker[2], Anja Ruud Winther[1], Xue Liu[2,3], Jan-Willem Veening[2], Davide Porcellato[1], Morten Kjos** [1]*

**1** Faculty of Chemistry, Biotechnology and Food Science, Norwegian University of Life Sciences, Ås, Norway, **2** Department of Fundamental Microbiology, University of Lausanne, Lausanne, Switzerland, **3** Guangdong Provincial Key Laboratory of Regional Immunity and Diseases, Department of Pathogen Biology, Shenzhen University Medical School, Shenzhen, Guangdong, China

* morten.kjos@nmbu.no

## Abstract

*Staphylococcus aureus* is a major concern in the dairy industry due to its significance as a pathogen causing bovine mastitis as well as a source of food poisoning. The nutrient-rich milk environment supports bacterial growth, but the specific genetic determinants that facilitate *S. aureus* proliferation and persistence in milk are poorly understood. In this study, we conducted a genome-wide CRISPR interference sequencing (CRISPRi-seq) screen with the laboratory strain *S. aureus* NCTC8325–4, to identify fitness determinants essential for *S. aureus* growth and survival in milk. We identified 282 milk-essential genes, including those with key roles in DNA replication, protein synthesis, and metabolism. Comparative analysis with brain heart infusion (BHI) as growth medium, revealed 79 genes with differential fitness, highlighting specific adaptations required for growth in milk. Notably, we found that purine biosynthesis, folate cycle pathways, and metal acquisition were particularly important in this environment. Based on this, we further demonstrate that *S. aureus* is more sensitive to the folate inhibitors trimethoprim-sulfamethoxazole (TMP-SMX) in milk and identify several genes whose knockdown results in hypersensitivity to TMP-SMX in milk. Additionally, our analysis showed a relatively reduced importance of cell wall components, such as teichoic acids, for *S. aureus* fitness in milk, which is also reflected in reduced efficiency of antimicrobials targeting teichoic acids. Together, these findings provide new insights into the genetic basis of *S. aureus* fitness and antibiotic susceptibility in milk, offering directions for novel treatment strategies against bovine mastitis.

## Author summary

*Staphylococcus aureus* is a leading cause of bovine mastitis, a condition that affects milk production and quality, with economic consequences for the dairy industry. The ability of *S. aureus* to survive in the milk environment is a key factor in its pathogenicity, yet many of the genetic factors that support its fitness in this niche remains unknown. By

**Data availability statement:** All sequencing data generated in this study are available on SRA (accession number PRJNA1173950).

**Funding:** This work was supported by a Joint Programming for Antimicrobial Resistance grant from Research Council of Norway (grant 296906) to M.K. and Swiss National Science Foundation (grant 40AR40_185533) to J.-W.V., and a FRIPRO grant from Research Council of Norway to D.P. (grant 314733). X.L. was supported by the National Key Research and Development Program of China (2023YFD1800100), the Science and Technology Project of Shenzhen (JCYJ20220818095602006), National Nature Science Foundation of China (82270012), and Shenzhen University 2035 Program for Excellent Research (86901-00000216). X.L. and J.-W.V. were supported by the Swiss National Science Foundation (SNSF) NCCR AntiResist 51NF40_180541. The funders had no role in study design, data collection and analysis, decision to publish, or preparation of the manuscript.

**Competing interests:** The authors have declared that no competing interests exist.

identifying these genetic determinants, this study sheds light on the mechanisms that enable *S. aureus* to persist in milk and highlights potential vulnerabilities that could be targeted to develop improved treatments. Understanding how *S. aureus* interacts with the milk environment is crucial for controlling infections in dairy cattle and preventing the spread of antibiotic resistance.

## Introduction

*Staphylococcus aureus* is known for causing a wide range of infections in both humans and animals. In the dairy industry, *S. aureus* is a significant cause of mastitis, an inflammation of the mammary gland which reduces milk quality and yield in dairy cattle [1]. This pathogen poses risks not only through intramammary infections but also by contaminating raw milk, where its ability to produce highly thermo-stable and protease-resistant enterotoxins can lead to foodborne illnesses [2, 3].

Milk is a nutrient-rich medium that supports the growth of bacteria, including *S. aureus*, due to the abundance of carbohydrates, fats, proteins, vitamins, and minerals [4, 5]. *S. aureus* efficiently hydrolyzes casein and ferments lactose, which further facilitates its proliferation in milk [6, 7]. Beyond promoting replication, the milk environment can also modulate the expression of various virulence factors that enhance immune evasion and persistence. These factors include modifications to cell surface characteristics, such as increased capsular polysaccharide expression, which reduces susceptibility to phagocytosis, and the secretion of toxins like leukocidins that target and kill bovine neutrophils [8–10]. Additionally, growth in milk has been shown to promote the expression of biofilm-associated proteins facilitating surface adherence and colonization, lactose permeases for improved lactose uptake as well as extracellular proteases that break down milk proteins to provide essential peptides and amino acids [11–14]. Finally, upregulation of genes involved in purine and lysine biosynthesis has also been reported for *S. aureus* grown in milk [11].

Current mastitis treatment relies on antibiotics, with β-lactams such as penicillin being used most commonly. However, as penicillin resistance becomes an increasing problem, other categories of antimicrobials – including aminoglycosides, lincosamides, macrolides, and sulfonamides – are employed, alongside combination therapies like amoxicillin-clavulanic acid (AMC) and trimethoprim-sulfamethoxazole (TMP-SMX) [15–18]. TMP-SMX acts synergistically by inhibiting sequential steps in the folate pathway, thereby disrupting nucleotide biosynthesis [19]. However, the efficacy of TMP-SMX in the milk environment is poorly understood. Furthermore, effectively managing *S. aureus*-associated bovine mastitis is challenging due to inconsistent antibiotic efficacy influenced by host-level factors such as age, somatic cell count, and number of infected quarters, as well as pathogen-level factors, such as antimicrobial resistance [1,20]. The dairy environment complicates treatment as variations in nutrient availability, biofilm production, and the milk's complex composition can alter bacterial susceptibility to antibiotics [20].

Despite existing studies on gene expression and regulation of *S. aureus* in milk, comprehensive knowledge on the genome-wide fitness determinants critical for *S. aureus* growth, survival and antibiotic susceptibility in this environment remains limited. This study aims to bridge this knowledge gap by exploring the genetic basis of *S. aureus* fitness in milk using a genome-wide CRISPR interference sequencing (CRISPRi-seq) screening approach [21,22]. CRISPRi relies on a catalytically dead Cas9 (dCas9) protein and gene-targeting single guide RNA (sgRNA), which together form a complex that bind to a gene of interest

and represses transcription [23]. By using a pooled sgRNA library that targets over 97% of the genetic features encoded by *S. aureus* [22], the Illumina-sequencing based CRISPRi-seq method enables precise knockdown of genes across the genome. This approach allows us to efficiently quantify how genes and pathways of *S. aureus* contribute to fitness in the milk environment. By identifying these fitness determinants, we aim to uncover potential vulnerabilities that could inform more effective treatment strategies against *S. aureus* infections in the dairy industry.

## Results

### Construction of a tetracycline-inducible CRISPRi system in *Staphylococcus aureus* NCTC8325–4

To enable genome-wide fitness quantification using CRISPRi-seq during growth of *S. aureus* strain NCTC8325–4 in milk, a stable and inducible CRISPRi system is essential. We initially evaluated the functionality of our two-plasmid, IPTG-inducible CRISPRi system, which has previously shown success in standard laboratory media [22]. To assess the functionality of the system, a CRISPRi strain targeting the essential gene *pbp1* was grown in ultra-high temperature (UHT) milk with and without IPTG induction and compared to a control strain harboring a non-targeting sgRNA. Colony-forming units per milliliter (CFU/ml) were determined at various time points. For the CRISPRi(*pbp1*) strain there was a clear reduction in growth compared to the control strain even without inducer (S1 Fig), indicating leaky expression of *dcas9* in milk. Possibly, this is due to partial induction of the IPTG-inducible promoter by the structural analog lactose present in milk.

To address this limitation, we constructed a strain where *dcas9* expression is controlled by a tetracycline-inducible promoter ($P_{tet}$) integrated into the chromosome of *S. aureus* NCTC8325–4 (*tetR*-$P_{tet}$-*dcas9*, Fig 1A). The $P_{tet}$ promoter allows induction by tetracycline (tet) or tetracycline-derivates such as anhydrotetracycline (aTc), and a similar system has been used in *S. aureus* JE2 [24]. Next, to test the functionality of the system in targeting an essential gene, the plasmid pVL2336-sgRNA(*pbp1*), which provides constitutive expression of the sgRNA targeting *pbp1*, was introduced into strain MM267. We initially validated the functionality of the CRISPRi system in brain heart infusion (BHI) medium, observing decreased growth of the *pbp1*-targeting strain only upon induction with aTc (concentrations between 7.5 and 120 ng/ml, Fig 1B). For the further experiments, 30 ng/ml aTc was used. It should be noted that prolonged incubation (>10 hours) led to a resurgence in growth of the inhibited strains, likely due to mutations in the CRISPRi regulatory system, consistent with observations in other studies [21]. We then tested the *pbp1*-knockdown strain in milk, with or without aTc induction (30 ng/ml), and measured CFU/ml at different time points. The *pbp1*-knockdown resulted in growth inhibition for the first eight hours of incubation (Fig 1C), and importantly, in the absence of inducer, the strain exhibited growth comparable to the control strain. This suggests minimal leakage from the $P_{tet}$ promoter under these conditions, confirming the suitability of this CRISPRi-system for use in milk.

### Quantifying the fitness determinants of *S. aureus* in milk by CRISPRi-seq

To identify genes essential for the growth of *S. aureus* in milk, we utilized our previously constructed sgRNA library targeting 1928 transcriptional units (TUs) of the *S. aureus* NCTC8325–4 genome (targeting > 97% of all genes) [22]. This library was transformed into *S. aureus* strain MM267 (NCTC8325–4, *tetR*-$P_{tet}$-dCas9, *tetM*), resulting in a tetracycline-inducible, pooled CRISPRi library in *S. aureus* NCTC8325–4 (Fig 2A). We conducted a CRISPRi-seq analysis in UHT milk, growing the CRISPRi library with or without aTc

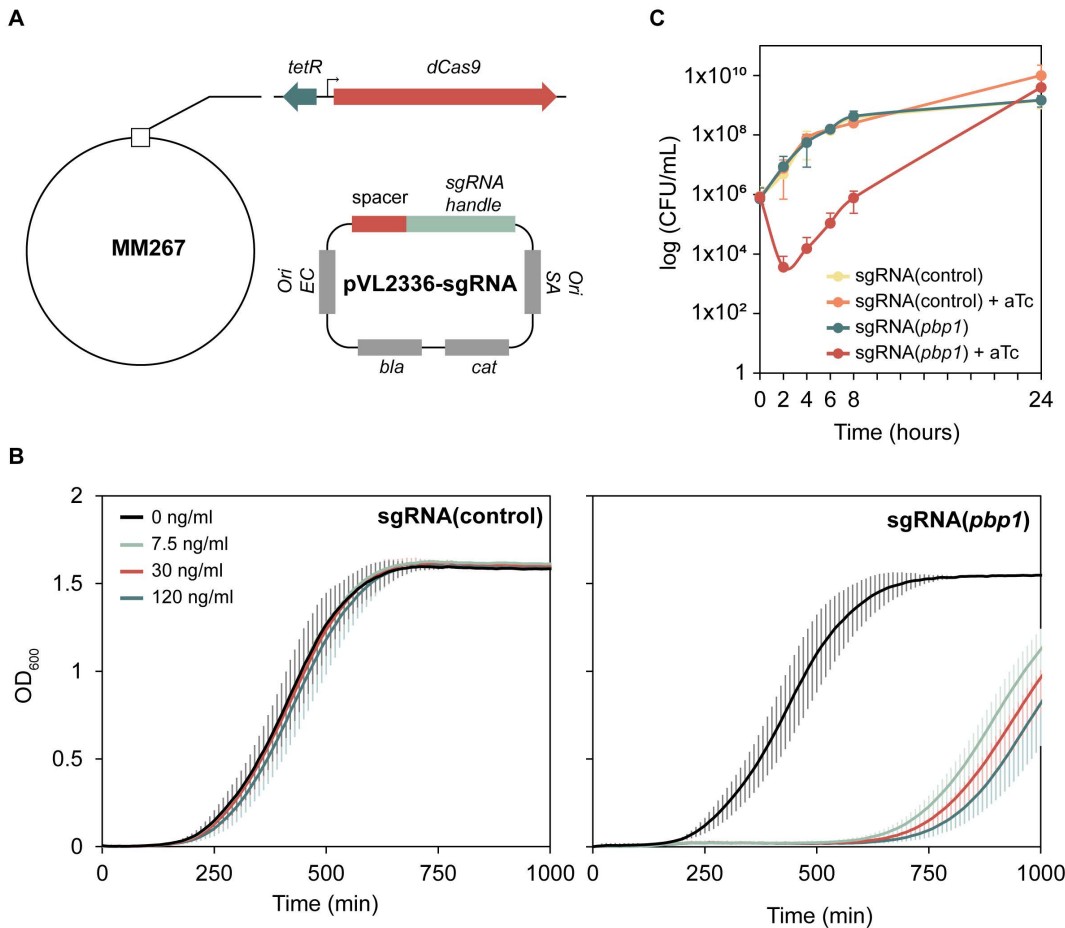

**Fig 1. (A)** Schematic representation of P$_{tet}$-*dcas9* on the *S. aureus* NCTC8325-4 chromosome, and the pVL2336-sgRNA (target) plasmid. **(B)** Verification of the tetracycline-inducible CRISPRi system in BHI. Growth curves of CRISPRi strains targeting *pbp1* (MM269) and a control strain harboring a non-targeting sgRNA (MM268) in BHI in the presence of varying concentrations of aTc, using a 4-fold dilution series from 7.5 ng/mL to 120 ng/mL. Optical density (OD$_{600}$) was measured at 10-min intervals. Data represent the averages of three independent experiments, each consisting of three technical replicates. Error bars represent the standard error based on these three independent experiments. **(C)** Verification of the tetracycline-inducible CRISPRi system in UHT milk. CRISPRi strains targeting *pbp1* (MM269) and a control strain harboring a non-targeting sgRNA (MM268) were grown in UHT milk with or without induction with 30 ng/ml aTc for 24 hours, and CFU/ml was calculated at 0, 2, 4, 6 and 24 hours. The data represent the average of two independent experiments, with error bars indicating standard error.

induction for ~20 generations at 37°C. The sgRNA abundances were determined by Illumina sequencing, as described previously [21] (Fig 2A). Principle component analysis (PCA) of the samples showed clear separation between induced and uninduced conditions (S2A Fig). Fitness score analysis identified 282 targets with a log$_2$ fold-change (L2FC) ≤ -1 with $p < 0.05$, and these were categorized as essential for growth in milk (S2B Fig). A full list of milk-essential genes is provided in S1 Table.

## Identifying milk-specific fitness determinants

Among the genes classified as essential in milk, many encode proteins involved in core cellular processes such as DNA replication, protein synthesis, cell division and central metabolic pathways; functions that are typically essential across various conditions. To filter out genes

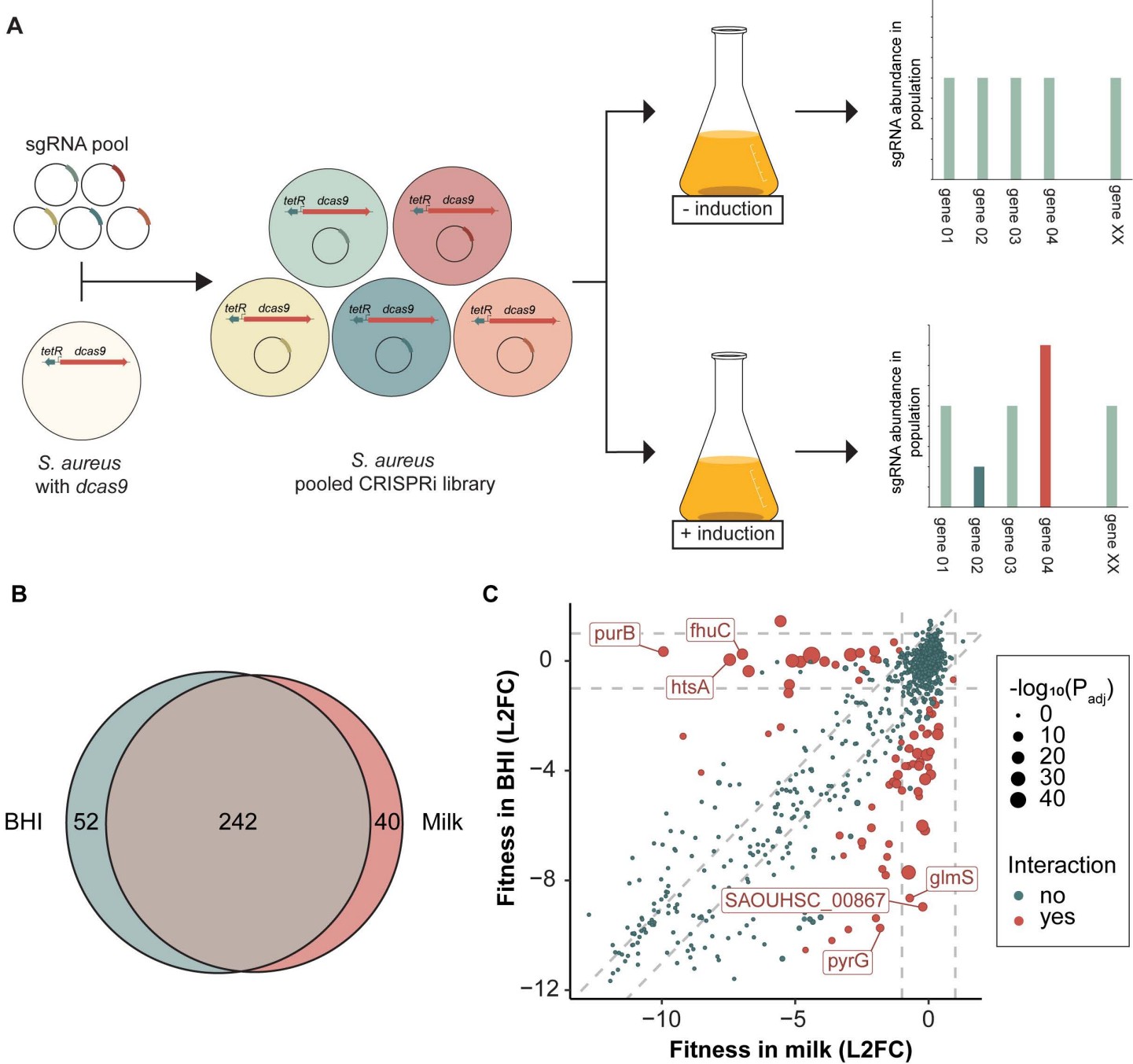

**Fig 2. (A)** Schematic overview of the construction of a CRISPRi pooled library and the subsequent CRISPRi-seq screen. **(B)** Venn-diagram showing the comparison of essential genes in BHI versus UHT milk identified by CRISPRi-seq. The Venn diagram was created using eulerr.co **(C)** Fitness effects upon *dcas9* induction in BHI versus milk. Genes with significant differential fitness effects ($|\Delta L2FC| > 1$, $P_{adj} < 0.05$) are depicted in red, with larger points indicating more statistically significant effects. The top three genes with the largest differential L2FC in both directions are marked with labels. Vertical and horizontal dashed grey lines indicate the $|L2FC| \geq 1$ threshold. Diagonal dashed grey lines represent the threshold where the difference in L2FC between BHI and UHT milk is greater than 1. Genes that fall far from these diagonal lines exhibit a larger differential fitness effect, highlighting conditionally essential genes. L2FC: $\log_2$ fold change.

specifically important for fitness in milk compared to another rich growth medium, we performed a similar CRISPRi-seq screen using the same library to quantify gene fitness in brain heart infusion (BHI) medium at 37ºC. Again, PCA showed clear separation between induced and uninduced conditions, as well as a separation between BHI and milk conditions (S2A Fig). The small separation between the uninduced conditions, suggest that there may be a minor leakiness of the $P_{tet}$ system during the experiments, although this was not observed in our initial tests (Fig 1). In BHI, we identified 294 targets as essential for growth (L2FC ≤ -1 with $p < 0.05$) (S2C Fig). Of these, 242 targets were classified as essential in both milk and BHI (Fig 2B). To further distinguish between these conditions, we compared the fitness effects between milk and BHI (Fig 2C). This analysis revealed 79 targets with significantly differential fitness in milk relative to BHI ($|\Delta L2FC| \geq 1$, $p < 0.05$). Specifically, 25 sgRNA targets were significantly more important for fitness in milk than in BHI, while 54 sgRNA targets were significantly less important for fitness in milk compared to BHI (Fig 2C and Tables 1 and S3).

**Table 1. List of genes more important for fitness in milk as determined by CRISPRi-seq.**

| Locus tag targeted (SAOUHSC) | Target gene(s)[a] | Function/pathway[b] | Interaction L2FC[c] | Interaction $P_{adj}$[c] |
|---|---|---|---|---|
| _02126 | *purB* | Purine/AMP biosynthesis | -10.5 | 1.3E-12 |
| _02430 | *htsABC* | Iron uptake (siderophore system) | -7.5 | 5.6E-21 |
| _00652 | *fhuCBG* | Iron uptake (siderophore system) | -7.2 | 3.4E-15 |
| _02139 | *pncA-ppaC* | Putative nicotinamidase - pyrophosphatase | -7.1 | 4.9E-18 |
| _01216 | *sucCD* | Succinyl-CoA synthetase, TCA cycle | -6.4 | 8.3E-19 |
| _01435 | *thyA* | Thymidylate synthase, dTMP synthesis | -5.7 | 5.0E-03 |
| _02158 | *aspB* | Aspartate transaminase | -5.1 | 6.4E-26 |
| _00019 | *purA* | Purine biosynthesis | -4.8 | 5.4E-19 |
| _01008 | *purEKCSQ-LFMNHD* | Purine biosynthesis | -4.6 | 9.7E-51 |
| _00637 | *mntABC* | Manganese ABC-transporter | -4.3 | 6.7E-14 |
| _00790 | *clpP* | Clp protease | -4.0 | 6.6E-11 |
| _00743 | *nrdF* | Ribonucleotide-diphosphate reductase | -4.0 | 3.6E-02 |
| _00274 | 00274 | Type VII secretion system, DUF600 | -3.9 | 3.6E-12 |
| _00269 | *esaG* | Type VII secretion system immunity | -3.3 | 5.7E-05 |
| _01007 | *folD* | 5,10-methylene-tetrahydrofolate dehydrogenase | -3.1 | 1.2E-25 |
| _00573 | *hemQ* | Heme peroxidase | -3.1 | 2.8E-02 |
| _00551 | 00551-*bshB2-folE2* | | -3.0 | 2.3E-04 |
| _01326 | 01326 | Amino acid permease | -2.9 | 9.9E-10 |
| _01064 | *pycA* | Pyruvate carboxylase | -2.4 | 1.3E-12 |
| _00957 | 00957 | TerC family integral membrane protein | -2.1 | 1.2E-03 |
| _00341 | *metICFE* | Methionine biosynthesis | -2.1 | 1.9E-04 |
| _00358 | 00358 | Integral membrane protein | -2.0 | 1.2E-04 |
| _00620 | *sarA* | Staphylococcal accessory regulator A | -1.9 | 7.1E-04 |
| _01908 | *smdA* | Cell morphology determinant | -1.9 | 1.3E-03 |
| _00542 | 00542-*azo1* | HAD superfamily hydrolase - FMN reductase | -1.5 | 3.7E-02 |

[a]Genes in the same operon as the target gene is also indicated.

[b]Functional characterization from AureoWiki [25].

[c]L2FC ($\log_2$ fold change) in fitness upon CRISPRi depletion and adjusted $p$-values ($P_{adj}$) from DESeq2 analysis.

## Purine and folate cycle metabolism are important for fitness during growth in milk

The CRISPRi-seq results highlighted that purine biosynthesis pathways were specifically important for *S. aureus* fitness in milk. Significant hits included the *purE* operon (*purEKCSQLFMNHD*, targeted by the *purE* sgRNA), along with the monocistronic *purA* and *purB* (Table 1 and Fig 3). The *purE* operon encodes enzymes that drive the conversion of phosphoribosyl pyrophosphate (PRPP) to inosine monophosphate (IMP), which is then converted to adenylosuccinate by *purA* and subsequently to AMP by *purB*. To validate these findings, we constructed individual CRISPRi strains with sgRNAs targeting *purA*, *purB* and *purE,* and compared their growth in BHI and milk to that of a control strain harboring a non-targeting sgRNA. The results showed that *purB* was particularly critical for growth in milk, also in this non-competitive setting, while *purA* and *purE* knockdowns exhibited only very mild growth defects (Fig 3A-B). Supporting the notion that purine biosynthesis is needed for optimal growth under these conditions, we show that supplementing the milk with the purine adenine indeed mitigates the dependency of purine biosynthesis as seen by the rescued the growth of the *purB* mutant under these conditions (Fig 3C).

The CRISPRi-seq results also emphasize the importance of directing precursors of the purine biosynthesis pathway, such as L-glutamine and ribulose-5-phosphate, towards purine biosynthesis rather than alternative pathways when *S. aureus* grow in milk. For example, depletion of *glmS* and *pyrG*, which utilize the *purF*-substate L-glutamine for peptidoglycan synthesis or L-glutamate synthesis, respectively, have less impact on fitness in milk compared to BHI (Fig 3D and S3 Table). The same trend is also seen for *pgcA* and *rpiA* which utilize ribulose-5-phosphate in alternative pathways (Fig 3D and S3 Table). Finally, we also observed that genes associated with the folate cycle (*folD*, *metEF*, *thyA*), which is intertwined with purine biosynthesis (Fig 3D), also appear to be critical for optimal fitness in milk (Fig 3B and Table 1).

## Increased sensitivity to trimethoprim-sulfamethoxazole in milk

Given that the results above point towards the importance of purine biosynthesis and the folate cycle for the fitness of *S. aureus* in milk, we explored the sensitivity of *S. aureus* to TMP-SMX under these conditions. TMP-SMX is a commonly used antibiotic combination inhibiting key enzymes in the bacterial folate pathway (TMP targets DfrA, while SMX targets FolP, Fig 3D), which is crucial for nucleotide biosynthesis and ultimately DNA replication [26]. Given the close links between the folate pathway and the enzymes of the purine pathway and folate cycle, we compared the sensitivity of *S. aureus* towards TMP-SMX in milk with the sensitivity in BHI. Indeed, our results demonstrated that milk sensitized *S. aureus* NCTC8325–4 as well as the two bovine mastitis isolates *S. aureus* RF122 and MOK023, to TMP-SMX treatment (Figs 4A and S3A.

## Blocking of nucleoside import sensitizes *S. aureus* to TMP-SMX

Recent studies have shown the potential of emerging TMP-SMX resistance in *S. aureus* [27], and targets or pathways that can be used to increase the sensitivity or counteract resistance to TMP-SMX are therefore of interest. We performed an additional CRISPRi-seq experiment where we grew the CRISPRi library in milk, both with and without induction, exposing it to supra-MIC concentrations of TMP (0.1 μg/ml) and SMX (0.5 μg/ml), which correspond to a 2-fold MIC concentration, and incubated for 13 hours. This CRISPRi-seq screen identified several genes with a significant reduction in fold change (L2FC <-1, $P_{adj}$ < 0.05) in the presence of TMP-SMX, suggesting that their knockdown sensitizes cells to TMP-SMX (Fig 4B). The

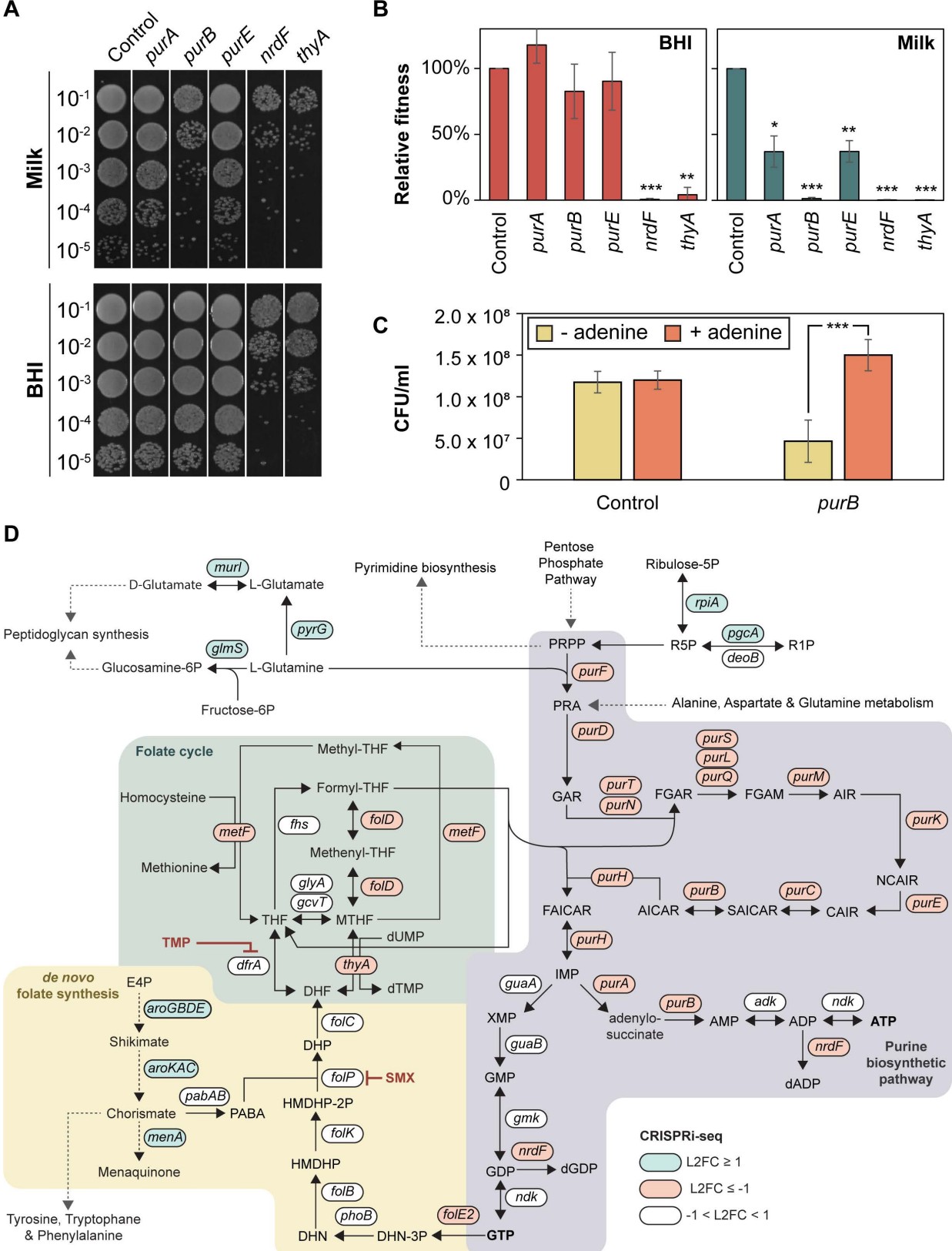

**Fig 3.** (A) Spotting assay of CRISPRi strains grown in UHT milk and BHI for ~ 20 generations in the presence of 30 ng/ml aTc. A 10-fold dilution series of each strain was prepared, and 5 μl of each dilution was spotted on BHI agar plates. The control strain harbors a non-targeting sgRNA. The assay was repeated twice with similar results. (B) CRISPRi strains grown in BHI (red) and UHT milk (green) for ~

20 generations in the presence of 30 ng/ml aTc. The relative fitness is defined as the percentage of CFU/ml of each strain relative to the control strain harboring a non-targeting sgRNA. Data represent the average and standard error based on three independent spotting assays. Statistical significance was determined using two-tailed t-tests comparing each strain against the control within each condition. Significance levels are indicated as follows: * $p < 0.05$, ** $p < 0.01$, *** $p < 0.001$. (C) CRISPRi strains grown in milk with (orange) or without (yellow) 10 μg/ml adenine. Data represent the average and standard error based on four independent spotting assays. Statistical significance was determined using two-tailed t-test. Significance levels are indicated as follows: * $p < 0.05$, ** $p < 0.01$, *** $p < 0.001$. (D). Overview of purine synthesis, folate synthesis, and the folate cycle in *S. aureus*. The figure depicts key metabolic pathways involved in the synthesis of purines and folate, and the interconnection of these pathways. Genes with differential $\log_2$ fold change (L2FC) in milk compared to BHI from the CRISPRi-seq screen are indicated in blue (L2FC ≥ 1, $p < 0.05$) or red (L2FC ≤ -1, $p < 0.05$). The purine biosynthetic pathway (highlighted in pink) details sequential conversion of phosphoribosyl pyrophosphate (PRPP) to inosine monophosphate (IMP) and subsequent generation of adenosine triphosphate (ATP) and guanosine triphosphate (GTP). GTP is connected to *de novo* folate biosynthesis, where it serves as a substrate in the production of 7,8-dihydroneopterin triphosphate (DHN-3P), an early precursor in the folate pathway. In the *de novo* folate biosynthesis pathway (highlighted in yellow), para-aminobenzoic acid (PABA) is synthesized through the Shikimate and Chorismate pathways and combines with 6-hydroxymethyldihydropterin pyrophosphate (HMDHP-2P) to generate dihydropteroate (DHP), which is converted to dihydrofolate (DHF). DHF is further processed within the folate cycle (highlighted in green), where it is reduced to tetrahydrofolate (THF), which then participates in the one-carbon pool essential for the biosynthesis of nucleotides, amino acids, and other key metabolites. The figure also highlights the inhibition points of trimethoprim (TMP) and sulfamethoxazole (SMX), which targets dihydrofolate reductase (DfrA) and dihydropteroate synthase (FolP), respectively. Abbreviations: PRPP: phosphoribosyl pyrophosphate; PRA: 5-phosphoribosylamine; GAR: 5'- phosphoribosylglycinamide; FGAR: 5'-phosphoribosyl-N-formylglycinamide; FGAM: 2-(formamido)-N1-(5'-phosphoribosyl)acetamidine; AIR: aminoimidazole ribotide; CAIR: 1-(5'-phosphoribosyl)-5-amino-4-imidazolecarboxylate; SAICAR: 1-(5'-phosphoribosyl)-5-amino-4-(N-succinocarboxamide)-imidazole; AICAR: 1-(5'-phosphoribosyl)-5-amino-4-imidazolecarboxamide; FAICAR: 1-(5'-phosphoribosyl)-5-formamido-4-imidazolecarboxamide; IMP: inosine monophosphate; AMP: adenosine monophosphate; ADP: adenosine diphosphate; dADP: deoxyadenosine diphosphate; ATP: adenosine triphosphate; XMP: xanthosine monophosphate; GMP: guanosine monophosphate; GDP: guanosine diphosphate; dGDP: deoxyguanosine diphosphate; GTP: guanosine triphosphate; DHN-3P: 7,8-dihydroneopterin triphosphate; DHN: 7,8-dihydroneopterin; HMDHP: 6-hydroxymethyl-7,8-dihydropterin; HMDHP-2P: 6-hydroxymethyldihydropterin pyrophosphate; DHP: dihydropteroate; DHF: dihydrofolate; THF: tetrahydrofolate; MTHF: 5,10-methylene tetrahydrofolate; dUMP: deoxyuridine monophosphate; dTMP: deoxythymidine monophosphate; TMP: trimethoprim; SMX: sulfamethoxazole; PABA: para-aminobenzoic acid.

hits did not include genes directly involved in the folate biosynthesis pathway, possibly due to the essentiality of this pathway in milk. We selected the seven of the most significant targets from the screen for further experiments (Fig 4B and Table 2). A complete overview of gene fitness upon TMP-SMX treatment is provided in S2 Table.

The most prominent hit from the CRISPRi-seq screen was the nucleoside permease *nupC*. An individual CRISPRi strain targeting *nupC* for knockdown demonstrated dramatically increased sensitivity of *S. aureus* to TMP-SMX in milk (Fig 4C-D). Although less prominent, another nucleoside permease, *nupG*, was found to have a similar effect in milk, but not BHI. TMP and SMX act synergistically to inhibit the production of dTMP, a vital precursor for DNA synthesis (Fig 3D). Thus, when available from the environment, thymidine can be imported by NupC and/or NupG to partially bypass the effect of TMP-SMX. This blocking of thymidine-import likely sensitizes cells to TMP-SMX in thymidine-rich environments (Fig 4E) [28].

For the remaining genes tested (*polA*, *noc*, *ung*, *murB*, SAOUHSC_01782, SAOUHSC_02121), individual CRISPRi-strains were also constructed. All these strains displayed increased sensitivity to TMP-SMX in milk. Four genes (*polA*, *noc*, *murB*, SAOUHSC_02121) also displaying increased susceptibility to TMP-SMX when knocked down in BHI, albeit indeed less strongly than in milk (Fig 4C-D). The hits included several proteins linked to DNA metabolism, such as *ung*, encoding uracil-DNA glycosylase involved in DNA repair, and *noc*, which encodes a protein known to affect both cell division through nucleoid occlusion in addition to DNA replication in *S. aureus* [29, 30]. Another target, *polA*, encodes DNA polymerase I, which is essential for DNA replication [29]. The yet uncharacterized genes SAOUHSC_02121 and SAOUHSC_01782 were also shown to result in increased sensitivity to TMP-SMX (Fig 4C-D). Given the involvement of the other targets in DNA metabolism, we

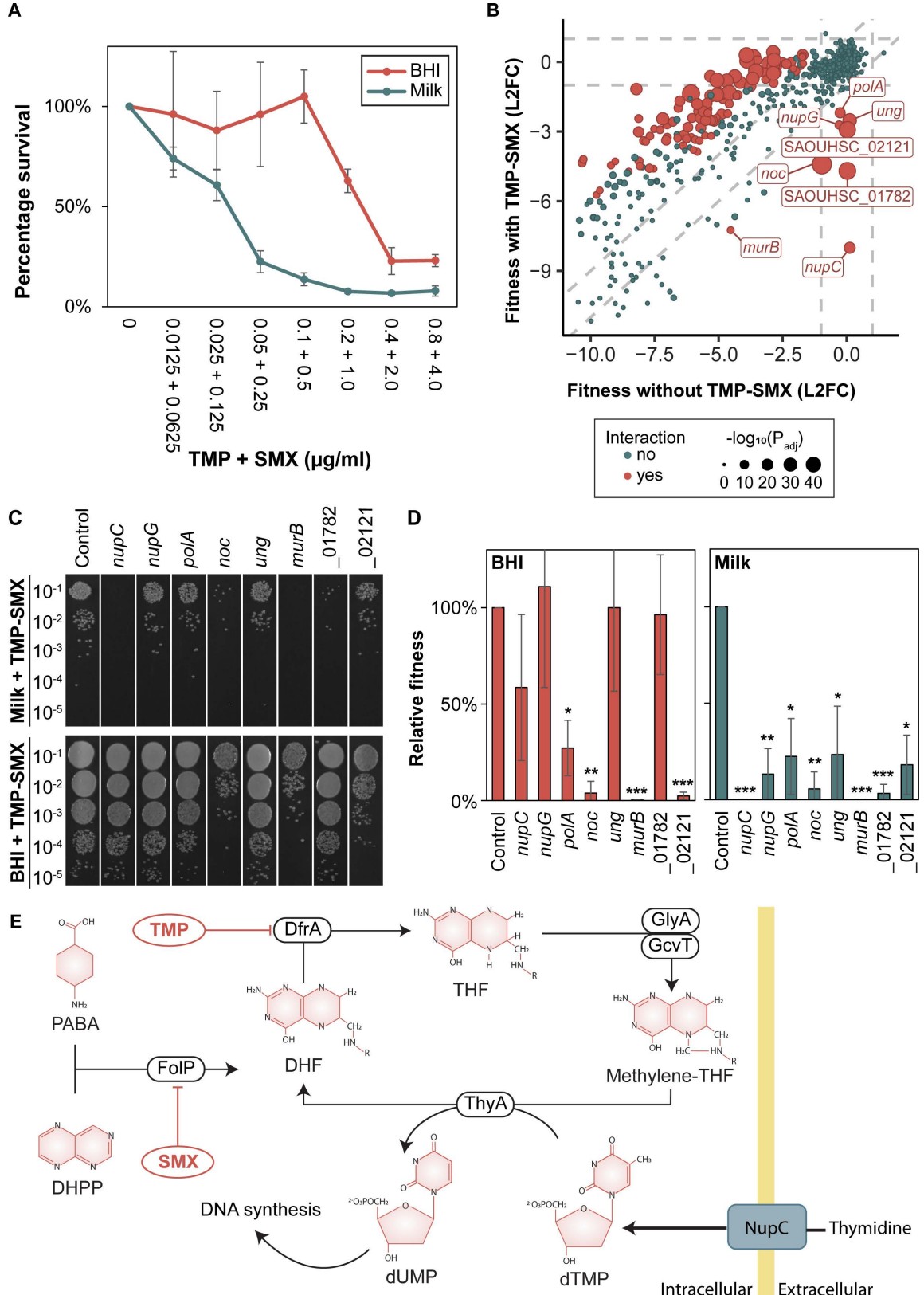

**Fig 4. (A) Increased sensitivity of *S. aureus* to trimethoprim-sulfamethoxazole (TMP-SMX) in milk compared to brain heart infusion (BHI) medium. Cultures of *S. aureus* NCTC8325-4 were diluted 1/1000 in BHI and milk containing a 2-fold dilution**

of TMP-SMX starting from 0.8 µg/ml TMP and 4.0 µg/ml SMX and subsequently incubated at 37°C for 6 hours. CFU/ml was calculated from 5 µl aliquots spotted on BHI agar plates for each condition, and percentage survival was calculated compared to untreated cultures. Data represent the mean of three independent experiments, with error bars indicating standard errors. (B) Fitness effects upon *dcas9* induction in milk with TMP-SMX versus without TMP-SMX. Genes with significant differential fitness effects ($|L2FC| \geq 1$, $P_{adj} < 0.05$) are depicted in red, with larger points indicating more statistically significant effects. Conditionally essential genes selected for follow-up experiments are marked with labels. Vertical and horizontal dashed grey lines indicate the $|L2FC| \geq 1$ threshold. Diagonal dashed grey lines represent the threshold where the absolute difference in L2FC between +/- TMP-SMX is greater than 1. Genes that fall far from these diagonal lines exhibit a larger differential fitness effect, highlighting conditionally essential genes. L2FC: $\log_2$ fold change. (C) Spotting assay of CRISPRi strains grown in UHT milk (top) or BHI (bottom) for ~ 20 generations in the presence of 0.1 µg/ml TMP and 0.5 µg/ml SMX. 30 ng/ml aTc was added for induction of dCas9 expression. A 10-fold dilution series of each strain was prepared, and 5 µl of each dilution was spotted on BHI agar plates. The control strain harbors a non-targeting sgRNA. (D) CRISPRi strains grown in UHT milk (green) and BHI (red). Relative fitness is defined as the percentage of CFU/ml of each strain relative to the control strain harboring a non-targeting sgRNA. Data represent the average and standard deviation based on three independent experiments. Statistical significance was determined using two-tailed t-tests comparing each strain against the control within the same condition. Significance levels are indicated as follows: * $p < 0.05$, ** $p < 0.01$, *** $p < 0.001$. (E) Schematic illustration of the folate pathway, illustrating the role of NupC in the process. See also legend to Fig 3 for abbreviations.

**Table 2. Selected genes resulting in increased sensitivity to TMP-SMX upon depletion.**

| Locus tag targeted (SAOUHSC) | Target gene(s)[a] | Function[b] | Interaction L2FC[c] | Interaction $P_{adj}$[c] |
|---|---|---|---|---|
| _03049 | noc | nucleoid occlusion protein | -2,8 | 4,66E-57 |
| _01782 | *01782-01781* | unknown | -3,4 | 3,71E-29 |
| _02121 | *02121* | unknown | -2,1 | 3,89E-25 |
| _00564 | *ung* | uracil-DNA glycosylase | -1,9 | 2,44E-13 |
| _00501 | *nupC* | nucleoside permease | -5,9 | 1,85E-08 |
| _00752 | *murB-00751* | UDP-N-acetylenolpyruvoyl-glucosamine reductase | -3,3 | 1,66E-07 |
| _01797 | *polA* | DNA polymerase I | -1,5 | 4,43E-05 |
| _00648 | *nupG* | nucleoside transport protein | -1,9 | 0,003 |

[a]Genes in the same operon as the target gene is also indicated.

[b]Functional characterization from *Aureo*Wiki [25].

[c]L2FC ($\log_2$ fold change) in fitness upon CRISPRi depletion and adjusted *p*-values ($P_{adj}$) from DESeq2 analysis.

speculate that SAOUHSC_02121 and SAOUHSC_01782 have related roles in the cell, but this awaits further investigation.

### Genes associated with metal homeostasis affect growth fitness in milk

In addition to purine biosynthesis, CRISPRi-seq revealed several virulence-associated genes that were important for fitness in milk (Table 1). These included the regulator SarA, metal acquisition mechanisms involved in iron scavenging (*htsA, fhuC*), heme synthesis (*hemQ*) and manganese uptake (*mntA*). Single CRISPRi-strains were constructed for these genes and assayed for their ability to grow in milk, showing that *fhuC, mntA* and *hemQ* were particularly critical for growth in milk (Fig 5A-B).

### Reduced importance of cell wall biosynthesis pathways for fitness in milk

The CRISPRi-seq analysis also identified several genes and pathways that are less critical for *S. aureus* fitness in milk compared to BHI. Notably, this included genes in the shikimate pathway and, intriguingly, teichoic acid biosynthesis (S3 Table). For example, knockdown of genes involved in glycolipid/lipoteichoic acid synthesis (*pgcA, gtaB, ugtP, ltaA*) was detrimental for growth in BHI but not in milk, while the early steps of wall teichoic acid synthesis (*tagO,*

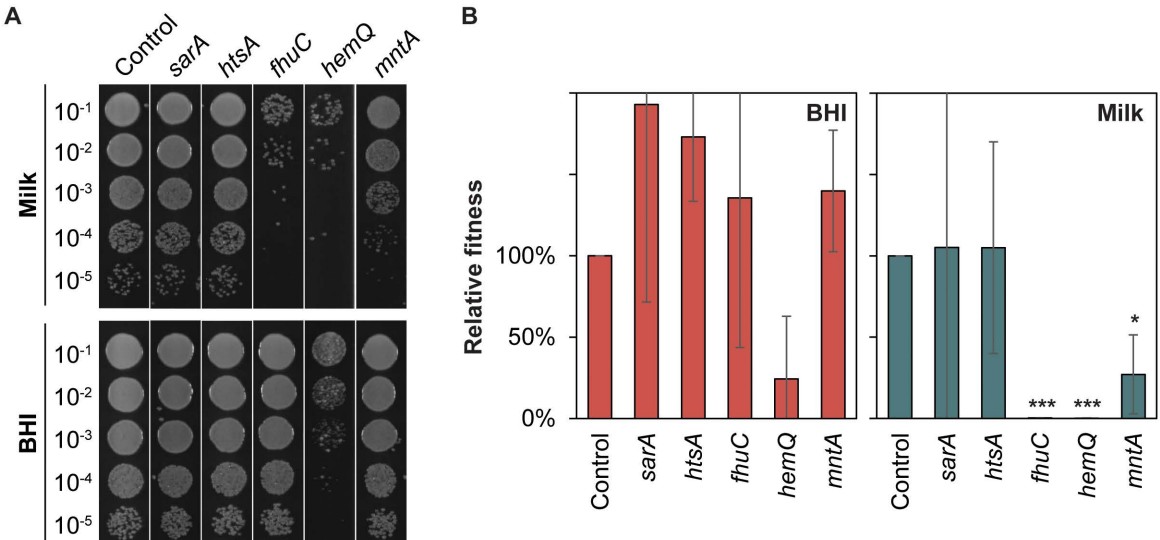

**Fig 5.** **(A) Spotting assay of CRISPRi strains grown in UHT milk (top) and BHI (bottom) for ~20 generations in the presence of 30 ng/ml aTc. A 10-fold dilution series of each strain was prepared, and 5 μl of each dilution was spotted on BHI agar plates. The control strain harbors a non-targeting sgRNA. (B) CRISPRi strains grown in BHI (red) and UHT milk (green) for ~ 20 generations in the presence of 30 ng/ml aTc. The relative fitness is presented as the percentage of CFU/ml of each strain relative to the control strain harboring a non-targeting sgRNA. Data represent the average and standard errors based on three independent experiments. Statistical significance was determined using two-tailed t-tests comparing each strain against the control within each condition. Significance levels are indicated as follows: * $p < 0.05$, ** $p < 0.01$, *** $p < 0.001$.**

*tagA*) also had a significantly reduced impact on fitness in milk compared to BHI, suggesting these genes are less critical for growth in the milk environment. The operon responsible for D-alanylation of teichoic acids, *dltABCD* (operon targeted at SAOUHSC_00867) was similarly less critical for growth in milk. We therefore hypothesized that the milk environment may affect the susceptibility to antimicrobials targeting teichoic acid biosynthesis. To test this, we examined the survival of cells treated with different concentrations of tunicamycin (targeting TagO) and targocil (targeting the wall teichoic acid exporter TarG) [31, 32]. Indeed, in both cases, growth in milk protected *S. aureus* NCTC8325–4 from the inhibitory effect of these compounds (Fig 6A-B). The sensitivity to tunicamycin was also tested for two bovine mastitis isolates, RF122 and MOK023 [33, 34], and the same observation was made for these (S3B Fig).

## Discussion

This study provides new insights into the genetic determinants of *S. aureus* fitness during growth in milk. Through a genome-wide CRISPRi-seq screen, we identified key mechanisms and pathways that play critical roles in *S. aureus* survival and proliferation in milk. Furthermore, we have identified genes involved in the susceptibility to the antibiotic combination TMP-SMX. Together, our findings have implications for control of *S. aureus* in the milk environment and the use of antimicrobial agents in treating *S. aureus*-associated bovine mastitis.

Our results highlight that the milk environment, although rich in nutrients, poses specific fitness challenges for *S. aureus*, which differ from those in standard laboratory media such as BHI. While several cellular processes, including DNA replication and protein synthesis, are required across both environments, purine biosynthesis and the folate cycle stood out as particularly critical for growth in milk (Table 1 and Fig 3). This finding aligns with previous studies showing that purine biosynthesis genes are upregulated in milk [11]. Purine biosynthesis is responsible for the production of purine nucleotides, essential for DNA and RNA

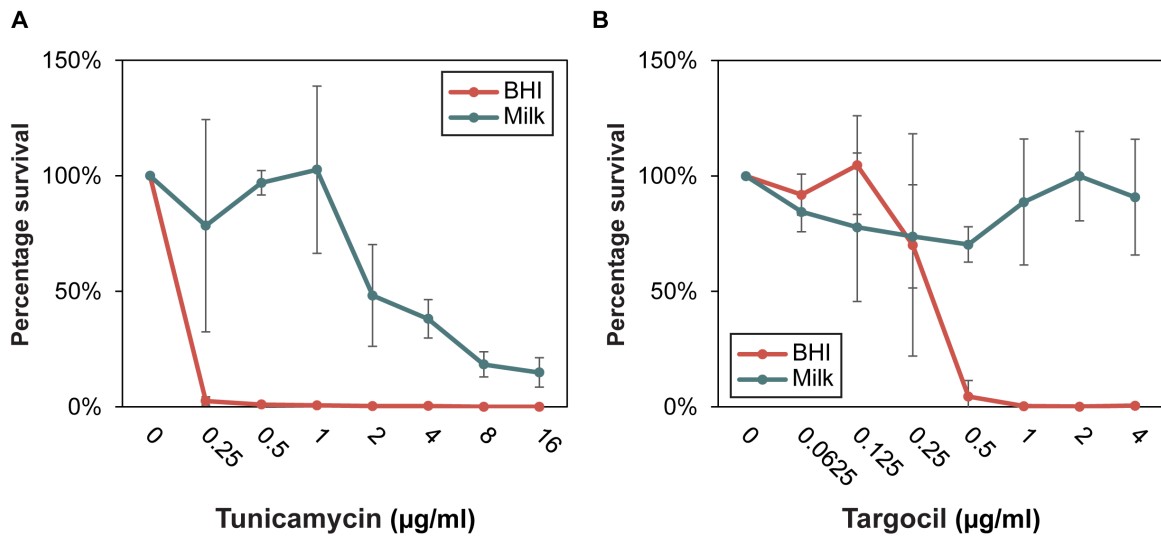

**Fig 6. Survival of *S. aureus* in response to tunicamycin (A) and targocil (B) treatment in BHI and milk.** Overnight cultures of *S. aureus* were diluted 1:1000 in BHI and milk, exposed to tunicamycin or targocil in a 2-fold dilution series, starting at 16 µg/ml and 4 µg/ml, respectively. Cultures were incubated at 37 °C for 6 hours before 10-fold dilutions were spotted onto BHI agar. *S. aureus* survival was calculated as the percentage of CFU/ml relative to untreated cultures. Data represent the average and standard errors based on three independent experiments.

synthesis, while the folate cycle provides one-carbon units needed for the biosynthesis of purines, thymine, and methionine (Fig 3D). The reliance on purine and folate metabolism is likely linked to the nutrient composition of milk, which may lack sufficient external sources of folate and purines for scavenging, forcing *S. aureus* to depend on its *de novo* biosynthetic pathways to sustain proliferation in milk. The importance of purine biosynthesis for *S. aureus* virulence is well established, and *de novo* purine biosynthesis has been shown to be required for the pathogenesis and virulence of *S. aureus* [35–39]. Our findings suggest that the milk environment may promote *S. aureus* to establish and maintain infection through the dependence of this pathway.

The antibiotic combination TMP-SMX inhibits synthesis of folates and thereby DNA synthesis (Fig 3D). TMP inhibits dihydrofolate reductase (DfrA), while SMX inhibits dihydropteroate synthase (FolP), both of which are crucial steps in the folate biosynthesis pathway. Our results demonstrate that *S. aureus* is clearly more sensitive to TMP-SMX in milk compared to BHI (Fig 4A). Interestingly, this finding contrasts with the observed reduced efficacy of penicillin G in the milk environment, which has been attributed to be due to the protective effect of milk components [40, 41] and the diminished activity of teichoic acid biosynthesis-targeting compounds observed in this study (Fig 6). These results underscore how the milk environment can both improve or reduce the efficiency of antimicrobials, depending on their mechanism. Specifically, the differential efficacy observed for TMP-SMX highlights the potential for using TMP-SMX more strategically in the milk environment, in cases of penicillin-resistant *S. aureus* or when penicillin efficacy is compromised.

To further explore the mechanisms underlying TMP-SMX susceptibility in milk, we performed a CRISPRi-seq screen, exposing the library to supra-MIC concentrations of TMP-SMX. The screen identified eight genes whose knockdown significantly increased sensitivity to TMP-SMX (Table 2 and Fig 4B-D). Our results suggest that *S. aureus* survival in the presence of TMP-SMX relies on environmental salvage of thymidine and/or other nucleosides, making

import of nucleosides via NupC and NupG essential in this setting. The screen also identified genes involved in DNA repair (*polA, noc, ung*) as essential, likely because TMP-SMX induces DNA replication stress, making DNA integrity and DNA repair pathways particularly critical for survival. These genes offer potential targets for novel treatments, either alone or in combination with TMP-SMX, to further sensitize *S. aureus*. Indeed, fluoroquinolones, which inhibit DNA replication, such as gemifloxacin, have already been shown to act synergistically with TMP-SMX against *S. aureus* [42].

Furthermore, our screen revealed several virulence-associated genes that are critical for fitness in milk, particularly those involved in metal acquisition. Uptake systems for iron (*htsABC, fhuC)* and manganese (*mntABC*) were essential for growth with knockdown significantly impairing bacterial fitness (Table 1). In milk, iron is limited due to sequestration by lactoferrin, transferrin, and casein, while manganese is restricted by calprotectin, both of which are part of the host's innate immune defense [43–47]. In general, the concentration of iron in milk ranges from 0.2 to 0.3 µg/ml, while manganese levels range from 0.02 µg/ml to 0.16 µg/ml [47]. In response, *S. aureus* relies heavily on high-affinity metal scavenging systems, using siderophores like staphyloferrin A and B to capture iron and transport it into the cell via HtsABC and SirABC, powered by the ATPase FhuC [43,44,48,49]. Similarly, manganese uptake is mediated by the MntABC and MntH transporters [43,50]. While *htsA* displayed significantly reduced fitness in the CRISPRi-seq screen, the single knockdown strain did not show a substantial reduction in growth (Fig 5A-B). This is likely due to the competitive nature of the pooled CRISPRi library, in which relatively mild fitness defects will be more easily detected compared to growth of single knockdown strains. For example, in the pooled library, other bacterial cells will take up and contribute to depletion of iron in the environment, thereby making it even harder for *htsA* knockdown strains to grow compared to pure culture of the same strain. Targeting iron and manganese acquisition systems represents a potential therapeutic strategy to limit *S. aureus* proliferation in the milk environment. Indeed, lactoferrin has shown inhibitory effects alone, and in combination with penicillin, on *S. aureus*, highlighting the potential for combination therapies [51–53].

Additionally, our screen indicated that the transcriptional regulator, SarA, was important for fitness of *S. aureus* in milk (Table 1). SarA plays a central role in regulating numerous processes, including the upregulation of fibronectin- and fibrinogen-binding proteins, hemolysins, toxins, oxidative stress response genes, and biofilm formation, while repressing expression of proteases, protein A, and collagen-binding proteins [54–57]. It also modulates the expression of other regulatory loci [57–60], making its overall role in bacterial fitness and virulence complex. Given the broad range of processes influenced by SarA, further investigation is needed to understand the specific functions of SarA in this environment.

In contrast to previous studies [14], we did not observe a contribution of secreted extracellular proteases to *S. aureus* fitness in milk. This is probably due to our experimental design using a pooled library. Here, extracellular proteases secreted to the medium will be public goods, possibly masking any effect of knockdown in individual strains. Additionally, we harvested cells in the exponential growth phase, and it is likely that secreted proteases contribute more during longer incubation periods when nutrient sources in the milk become depleted, as suggested earlier [14]. It should also be noted that the high temperature treatment of the milk used in our experiments, probably results in changes in the protein composition which may also influence the results [61].

Interestingly, due to the use of CRISPRi-seq, we could also identify pathways that are essential in normal growth medium, but redundant or less important for fitness in milk, (S3 Table and Fig 3D). This includes the Shikimate pathway (e.g., *aroA, aroA2, aroB, aroK, aroE*

as well as *menA*) essential for the synthesis of aromatic amino acids and menaquinone (vitamin K). This suggests that *S. aureus* acquires these metabolites from the milk environment. As a result, targeting the Shikimate pathway, which has been proposed as an antimicrobial target [62, 63], may be less effective in treating milk-associated infections like mastitis. Additionally, genes involved in teichoic acid and glycolipid biosynthesis, such as *pgcA, gtaB,* and *ugtP-ltaA*, which are required for growth in BHI, were found to be non-essential in milk (S3 Table). Furthermore, TagO and TagA, responsible for early steps in wall teichoic acid synthesis, had significantly reduced fitness effects in milk compared to BHI (S3 Table). We also noted that *vraX*, which encodes a cell wall stress factor [64], was less important for growth in milk, along with other cell-wall-associated genes such as *sepF, lyrA* and *rapZ* (SAOUHSC_00787). These findings suggest that the cell wall architecture of *S. aureus* may be less critical for survival in milk, potentially due to the protective nature of the milk environment, where *S. aureus* can associate with milk fat globules [40]. This also raises questions about the efficacy of antibiotics targeting cell wall biosynthesis in mastitis treatment. Indeed, our results indicate that growth in milk confers protection against the cell wall-targeting antibiotics tunicamycin and targocil (Fig 6A-B). Similarly, the observed reduced efficacy of penicillin G in the milk environment has been attributed to the protective effect of milk components, which may shield *S. aureus* from the action of the antibiotic [40, 41].

A key limitation of this study, is that most experiments were done with the *S. aureus* strain NCTC8325–4, a laboratory strain, originally isolated from human sepsis, lacking prophages and characterized by harboring mutations resulting in reduced activity of the stress sigma factor SigB [65]. While we do not know how representative NCTC8325–4 is for bovine mastitis derived strains, the observed effects of TMP-SMX and tunicamycin on NCTC8325–4 in milk was mirrored in the bovine isolates tested (RF122 and MOK023) (S3 Fig). It is also worth noting that, *sigB*-deficiency has been associated with chronic udder infections [66]. Furthermore, we also tested some of the key phenotypes related to knockdown of purine biosynthesis (*purB*) and iron uptake (*fhuC*) in *S. aureus* in the SH1000, a strain with a repaired SigB-response. The effects observed in SH1000 were similar as in NCTC8325–4 (S4 Fig), suggesting that *sigB* does not have major impact on the phenotype under the conditions tested here.

In summary, our study provides important insights into the genetic factors that influence *S. aureus* survival and growth in milk. By identifying milk-specific determinants and linking them to antibiotic susceptibility, we offer potential new avenues for improving the treatment of mastitis. Moreover, our findings highlight the importance of considering the unique properties of the milk environment when developing antimicrobial strategies, as pathways that are essential in other media can be less critical in milk, and vice versa. Future studies could build on our findings by applying this system to other infection-relevant conditions, or to different *S. aureus* strains isolated from bovine mastitis cases to explore strain-specific fitness determinants associated with virulence in the milk environment.

## Materials and Methods

### Bacterial strains and growth conditions

Bacterial strains used in this study are listed in S4 Table. *S. aureus* strains were grown at 37 °C in BHI medium with shaking, or on BHI agar plates. When needed, 5 μg/ml erythromycin and/or 10 μg/ml chloramphenicol were added for selection. Transformation was performed with electroporation using plasmid DNA as described before [67]. *E. coli* was grown in LB medium at 37 °C with shaking or on LA agar. Ampicillin (100 μg/ml) was added for selection. For transformation of *E. coli*, chemically competent cells were prepared using calcium chloride treatment, followed by transformation with heat shock according to standard protocols.

## Plasmid and strain construction

All plasmids used in this work are listed in S5 Table. Primers used for cloning are listed in S6 Table. Plasmid constructs were verified by PCR and sequencing.

**Construction of pMAD-tetR-Ptet-dcas9 for chromosomal integration.** The plasmid pMAD-*tetR*-Ptet-*dcas9* was constructed by Golden Gate cloning. The region upstream and downstream a neutral locus used for integration (between SAOUHSC_03046 and SAOUHSC_03047) was amplified with primer pairs mm40/mm32 and mm35/mm36, respectively, using chromosomal DNA of strain NCTC8325–4 as template. The *tetR*-Ptet-*dcas9* fragment was amplified from pFD152 [68] using primers mm33/mm34. All primers were designed to include BsaI restriction sites. Each PCR product was digested with BsaI-HF (New England Biolabs) and ligated into pMAD-GG using T4 DNA ligase (New England Biolabs) at a 3:1 insert-to-vector molar ratio. The ligation mix was transformed into *E. coli* IM08B. The resulting plasmid was verified by Sanger sequencing and transformed into electrocompetent *S. aureus* NCTC8325–4. Chromosomal integration of the plasmid was performed using the temperature-sensitive pMAD system., following the protocol described previously [69].

**Construction of sgRNA plasmids.** Individual sgRNA plasmids were constructed by cloning a 20-bp sgRNA sequence into pVL2336 using Golden Gate cloning as described before [22]. Briefly, forward and reverse oligos for each sgRNA (S6 Table) were diluted in TEN buffer (10 mM Tris, 1 mM EDTA, 100 mM NaCl, pH8), incubated at 95 °C for 5 minutes and then slowly cooled down to room temperature for annealing. The vector, pVL2336, was digested with BsmBI. The purified vector and the annealed oligos were ligated using T4 ligase and transformed into *E. coli* IM08B. Correct vectors were verified by Sanger sequencing.

## Construction of a pooled CRISPRi library in *S. aureus* NCTC8325–4

Plasmid isolated from the *E. coli* sgRNA library [22] were transformed into *S. aureus* MM267 (*S. aureus* NCTC8325–4, *tetR*-Ptet-*dcas9*, *tetM*) by electroporation. Multiple parallel transformation reactions were plated onto BHI agar with chloramphenicol selection. More than 250,000 colonies were collected using a cell scraper, pooled and resuspended in BHI with chloramphenicol selection. The pooled library was diluted to an $OD_{600}$ of 0.3 in BHI with chloramphenicol selection and re-grown at 37°C to an $OD_{600}$ of 0.8 and stored as glycerol stocks for further use.

## CRISPRi-seq screens

CRISPRi-seq experiments were performed following the protocol described previously [22] with some modifications. Briefly, the library was grown in 100 ml BHI medium or UHT milk (Tine SA, Oslo, Norway) supplemented with chloramphenicol for selection. When appropriate, 30 ng/ml aTc was added for CRISPRi induction. Four replicates per condition were used. The library was inoculated 1/1,000 from the stock, grown at 37 °C with shaking for 7 hours, then reinoculated 1/1,000 in fresh BHI or UHT milk, and grown for 6 additional hours (resulting in ~ 20 generations). For the TMP-SMX experiment, the same conditions were used, except that TMP and SMX were added to a final concentration of 0.1 µg/ml and 0.5 µg/ml, respectively.

For plasmid isolation from BHI, cells from 5 ml culture were harvested by centrifugation (4,000 x *g*, 5 min, 4°C) and lysed by treatment with lysozyme (0.8 mg/ml) and lysostaphin (40 µg/ml) at 37 °C for 30 minutes, followed by extraction using the E.Z.N.A. Plasmid Mini Kit I. For plasmid isolation from milk, cells were harvested by centrifugation (7,000 x *g*, 5 min, 4°C). The supernatant and any remaining fat were removed, and the pellet was washed in 20% (w/v)

sodium citrate until the supernatant was clear, before lysis and extraction in the same manner as samples from BHI.

### Library preparation and Illumina sequencing

For all CRISPRi-seq experiments, the Illumina amplicon library was prepared following previously described protocols [70]. Briefly, 2 µl of the plasmid was used as the template and amplified using Nextera DNA Indexes (Illumina Inc., San Diego, CA, USA) index primers. The PCR products were then purified and normalized using the SequalPrep Normalization Plate Kit (Thermo Fisher Scientific, Waltham, MA, USA). The purified libraries were quantified using the KAPA library quantification kit (KAPA Biosystems, Wilmington, MA, USA). The CRISPRi-seq experiments in milk and BHI (without TMP-SMX) were sequenced on an Illumina Miseq platform (Illumina Inc.) using the Miseq Reagent Kit v3 (Illumina Inc.), while the CRISPRi-seq with TMP-SMX was sent to Novogene (UK) for sequencing on the NovaSeq platform using 150 bp paired-end reads.

### CRISPRi-seq differential enrichment analyses

CRISPRi-seq differential enrichment analyses were done as previously described, with some modifications [22]. The read counts for each sgRNA were extracted using 2FAST2Q (v2.3.4) [71], with feature start position in the read set to 54. Otherwise, 2FAST2Q was used with default settings. Enrichment or depletion of sgRNAs was analyzed using DESeq2 [72], testing against an absolute L2FC of 1 with a significance threshold of $\alpha = 0.05$. For PCA analysis, DESeq2's blind rlog transformation was used for count normalization. Whenever L2FC are visualized or reported, these are shrunk with the apeglm method [73].

To account for variation in CRISPRi induction levels between TMP-SMX treated and untreated samples, we included a linear correction based on the number of generations of CRISPRi induction in the DESeq2 model. Specifically, instead of using of a binary treatment variable (induction yes/no), we included a numeric variable representing the degree of induction, by scaling and centering the number of generations of induction per treatment with the built-in R function scale(). The generation time was estimated based on CFU/ml in milk with and without TMP-SMX treatment, using 12 generations as the overall estimate for TMP-SMX treated samples, and 20 for untreated ones.

### Growth and antibiotic susceptibility assays

**Growth curves in 96-well microtiter plates.** For growth in 96-well microtiter plates, overnight cultures were diluted 1/1,000 in fresh BHI. Then, 300 µl of the diluted culture was added to each well in triplicates. $OD_{600}$ was measured every 10 minutes using either the Synergy H1 Hybrid Reader (BioTek) or the Hidex Sense (Hidex Oy), with 3 seconds shaking prior to each measurement. Where appropriate, chloramphenicol was added for selection, and aTc was added at 30 ng/ml for induction.

**Testing of CRISPRi systems in milk.** CFU/ml was calculated to assess the functionality of the CRISPRi systems in milk. Overnight cultures were diluted 1/1,000 in fresh milk and incubated at 37 °C with shaking. When appropriate, erythromycin and/or chloramphenicol was added for selection, and either 30 ng/ml aTc or 500 µM IPTG was added for induction. At specific timepoints, 100 µl of 10-fold serial dilutions were plated on BHI agar containing erythromycin and/or chloramphenicol. Plates were incubated overnight at 37 °C, and CFU/ml was calculated.

**Susceptibility testing for TMP-SMX, targocil and tunicamycin.** For susceptibility testing to TMP-SMX, targocil, and tunicamycin, overnight cultures of wild-type *S. aureus* were diluted 1/1,000 in BHI and milk. Cultures were exposed to the antibiotics in a 2-fold

dilution series and incubated at 37 °C for 6 hours. After incubation, 10-fold serial dilutions were prepared, and 5 μl of each dilution was spotted onto BHI agar. Plates were incubated overnight at 37 °C, and CFU/ml was calculated.

**Growth of individual CRISPRi strains.** For the growth of individual CRISPRi strains, overnight cultures were diluted 1/1,000 in 5 ml BHI and milk and incubated at 37 °C with shaking. After 7 hours of incubation, cultures were re-diluted 1/1,000 in fresh medium and incubated for an additional 6 hours, allowing for approximately 20 generations of growth. Chloramphenicol was added for selection. When appropriate, aTc (30 ng/ml) was added for induction and/or the media were supplemented with 10 μg/ml adenine (Sigma). TMP (0.1 mg/ml) and SMX (0.5 mg/ml) was included when necessary for susceptibility testing. After incubation, 10-fold serial dilutions of each strain were prepared, and 5 μl of each dilution was spotted on BHI agar. Plates were incubated overnight at 37 °C, and CFU/ml was calculated.

All growth and antibiotic susceptibility assays were repeated at least three times.

## Supporting information

**S1 Fig. CFU/ml of CRISPRi strains targeting *pbp1* (MK1857) and a control strain harboring a non-targeting sgRNA (MM75).** Strains were grown in UHT milk for 6 hours with or without induction with 500 μm IPTG. CFU/ml was calculated at 2-hour intervals.
(TIF)

**S2 Fig.** (A) Principal component analysis (PCA) of the rlog-transformed sgRNA counts. **(B)** Fitness effect upon dCas9 induction in UHT milk. **(C)** Fitness effect upon dCas9 induction in BHI. Genes with significant fitness effects ($|\log2FC| \geq 1$, *Padj* < 0.05) are depicted in red, while green points indicate neutral genes. Dashed grey lines denote the thresholds for significance ($|\log2FC| \geq 1$, *Padj* < 0.05). The data were analyzed using DESeq2 [72], with normalized counts calculated using the 'counts()' function, and log2FC values were shrunk using the 'apeglm' method [73].
(TIF)

**S3 Fig. Sensitivity of *S. aureus* RF122 (left) and MOK023 (right) to (A) trimethoprim-sulfamethoxazole (TMP-SMX) and (B) tunicamycin in milk compared to brain heart infusion (BHI).** Cultures of *S. aureus* were diluted 1/1000 in BHI and milk containing a 2-fold dilution of antibiotics and subsequently incubated at 37 °C for 6 hours. CFU/ml was calculated from 5 μl aliquots spotted on BHI agar plates, and percentage survival was calculated compared to untreated cultures. Data represent the mean of three independent experiments, with error bars indicating standard errors.
(TIF)

**S4 Fig. Relative fitness of *S. aureus* SH1000 CRISPRi strains grown in BHI (red) and UHT milk (green) for ~ 20 generations in the presence of 500 μM IPTG.** CFU/ml was calculated based on three independent spotting assays, and the relative fitness is presented as the percentage of CFU/ml of each strain relative to the control strain harboring a non-targeting sgRNA. Data represent the average and standard errors based on three independent experiments. Statistical significance was determined using two-tailed t-tests comparing each strain against the control within each condition. Significance levels are indicated as follows: * *p* < 0.05, ** *p* < 0.01, *** *p* < 0.001.
(TIF)

**S1 Table. CRISPRi screen to identify genes influencing fitness in milk versus BHI.** Log2 fold change (L2FC) and the adjusted p-value (padj) and the conclusion about significance ("essential" or "costly" if $|L2FC| \geq 1$, $P_{adj}$ 0.05) is shown for each sgRNA. Interaction effect

is also indicated, representing the differential effect of aTc induction in milk vs. BHI, and whether this is significant ("yes" if $|L2FC| \geq 1$, $P_{adj} < 0.05$).
(XLSX)

**S2 Table. CRISPRi screen to identify genes influencing TMP-SMX susceptibility.** Log2 fold change (L2FC) and the adjusted p-value (padj) and the conclusion about significance ("essential" or "costly" if $|L2FC| \geq 1$, $P_{adj}$ 0.05) is shown for each sgRNA in treated and untreated samples. Interaction effect is also indicated, representing the differential effect of aTc induction in the presence of TMP-SMX compared to untreated samples, and whether this is significant ("yes" if $|L2FC| \geq 1$, $P_{adj} < 0.05$).
(XLSX)

**S3 Table. List of genes less important for fitness in milk as determined by CRISPRi-seq.**
(DOCX)

**S4 Table. Strains used in this study.**
(DOCX)

**S5 Table. Plasmids used in this study.**
(DOCX)

**S6 Table. Oligos used in this study.**
(DOCX)

## Acknowledgments

We thank Orla Keane, Teagasc, Ireland for providing bovine clinical isolates. We also thank the reviewers for the helpful comments to the manuscript.

## Author contributions

**Conceptualization:** Marita Torrissen Mårli, Xue Liu, Jan-Willem Veening, Davide Porcellato, Morten Kjos.

**Formal analysis:** Marita Torrissen Mårli, Vincent de Bakker, Davide Porcellato.

**Funding acquisition:** Jan-Willem Veening, Morten Kjos.

**Investigation:** Marita Torrissen Mårli, Anne Olsdatter Ohren Nordraak, Anja Ruud Winther.

**Methodology:** Vincent de Bakker, Xue Liu, Jan-Willem Veening, Davide Porcellato, Morten Kjos.

**Project administration:** Davide Porcellato, Morten Kjos.

**Resources:** Morten Kjos.

**Supervision:** Jan-Willem Veening, Davide Porcellato, Morten Kjos.

**Visualization:** Marita Torrissen Mårli.

**Writing – original draft:** Marita Torrissen Mårli, Morten Kjos.

**Writing – review & editing:** Anne Olsdatter Ohren Nordraak, Vincent de Bakker, Anja Ruud Winther, Xue Liu, Jan-Willem Veening, Davide Porcellato.

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
