## [Decision Letter · Decision Letter 0]

3 Dec 2024

PPATHOGENS-D-24-02369

Genome-wide analysis of fitness determinants of Staphylococcus aureus during growth in milk

PLOS Pathogens

Dear Dr. Kjos,

Thank you for submitting your manuscript to PLOS Pathogens. After careful consideration, we feel that it has merit but does not fully meet PLOS Pathogens's publication criteria as it currently stands. The reviewers raises serious concerns about the choice of the S. aureus strain, which bears a sigB mutation and is not representative for bovine-adapted S. aureus. Also, the quantification of purine and folate in milk should be adressed and the analysis and presentation of data should be imporoved. Therefore, we invite you to submit a revised version of the manuscript that addresses the points raised during the review process.

Please submit your revised manuscript within 60 days Feb 01 2025 11:59PM. If you will need more time than this to complete your revisions, please reply to this message or contact the journal office at plospathogens@plos.org. Please include the following items when submitting your revised manuscript:

We look forward to receiving your revised manuscript.

Kind regards,

Andreas Peschel, Ph.D.

Academic Editor

PLOS Pathogens

Michael Otto

Section Editor

PLOS Pathogens

Michael Malim

Editor-in-Chief

PLOS Pathogens

orcid.org/0000-0002-7699-2064

**Journal Requirements:**

1) We do not publish any copyright or trademark symbols that usually accompany proprietary names, eg ©,  ®, or TM  (e.g. next to drug or reagent names). Therefore please remove all instances of trademark/copyright symbols throughout the text, including:

- ® on page: 29.

3) Please amend your detailed Financial Disclosure statement. This is published with the article. It must therefore be completed in full sentences and contain the exact wording you wish to be published.

1) State what role the funders took in the study. If the funders had no role in your study, please state: "The funders had no role in study design, data collection and analysis, decision to publish, or preparation of the manuscript.".

If you did not receive any funding for this study, please simply state: u201cThe authors received no specific funding for this work.

4) Please ensure that the funders and grant numbers match between the Financial Disclosure field and the Funding Information tab in your submission form. Note that the funders must be provided in the same order in both places as well. Currently, these funds "Key Research and Development Program of China (2023YFD1800100) and  Shenzhen University 2035 Program for Excellent Research (86901-00000216)" are missing from the Funding Information tab.

Please indicate by return email the full and correct funding information for your study and confirm the order in which funding contributions should appear. Please be sure to indicate whether the funders played any role in the study design, data collection and analysis, decision to publish, or preparation of the manuscript.

**Reviewers' Comments:**

Reviewer's Responses to Questions

**Part I - Summary**

Reviewer #1: The manuscript investigates the genetic fitness determinants of Staphylococcus aureus during growth in milk, employing a CRISPRi-seq system with the novel addition of aTc inducibility to enable its use in milk-based growth conditions. The study identifies 79 genes critical for growth in milk compared to BHI and shows pathways such as the purine cycle, folate biosynthesis, and metal acquisition to be of relevance. The authors validate the CRISPRi-seq dataset using individual knockdown strains and demonstrate an increased susceptibility of S. aureus to TMP-SMX during milk growth, emphasizing the importance of folate biosynthesis in this nutrient environment.

While this work is compelling and aligns well with the journal's scope, some significant shortcomings should be addressed before publication.

Reviewer #2: I find this to be an exceptionally clearly and well written paper. The authors use the relatively new technique of CRISPRi-seq to identify Staphylococcus aureus fitness genes involved in growth in milk. They found that purine biosynthesis and other genes were essential in the milk environment compared to growth in Brain Heart Infusion medium, whereas genes involved in teichoic acid biosynthesis were less important in the milk environment. S. aureus grown in milk was shown to be more susceptible to folate biosynthesis inhibitors - trimethoprim-sulfamethoxazole. Overall, the findings may inform antibiotic therapy of mastitis.

Reviewer #3: In this study, Marli and colleagues use a genome-wide CRISPR interference sequencing (CRISPRi-seq) screen to identify genes required for growth of Staphylococcus aureus in bovine milk. The strengths of the study include the genome-wide approach to identify pathways essential for growth in milk in comparison to nutrient-rich conditions (BHI).

Another strength is the application of these data to test therapeutic potential of inhibitors of the pathways identified.

These experiments indicated that trimethoprim-sulfamethoxazole has enhanced activity for inhibiting S. aureus growth in milk.

The authors also identify that teichoic acids may have lesser importance for growth in milk compared to BHI.

Overall, the study seems well carried out with appropriate controls and provides new information relating to S. aureus growth in milk.

However, the major weakness of the study is the use of the laboratory strain 8325-4 which is a UV-treated phage-cured derivative of a human clinical isolate from an eye infection. It is unclear why the authors did not use S. aureus clinical isolates from the established bovine-associated lineages responsible for mastitis worldwide.

Bovine S. aureus lineages are adapted to the dairy niche and specialised for causing disease in dairy cattle. Therefore, the strain selected is not representative of the strains that would be found in that niche. This limits the potential for identifying genes/pathways required for survival/growth in milk.

It is feasible that some of the genes/pathways identified in strain 8325-4 to be required for growth in milk are also important for growth in bovine S. aureus lineages but this has not been examined.

Furthermore, the therapeutic potential of targeting pathways required for growth in milk has not been tested in clinically-relevant bovine S. aureus isolates.

**Part II – Major Issues: Key Experiments Required for Acceptance**

Reviewer #1: 1. Relevance of Purine Cycle and Folate Biosynthesis in Milk:

o The authors highlight the importance of the purine cycle and folate biosynthesis during growth in milk, suggesting these nutrients are limiting in this environment. This is unexpected, as milk is nutritionally very rich.

o The authors might quantify purine and folate levels in milk, including both UHT milk (as used in the study) and fresh milk. Such data would clarify whether these nutrients are indeed limited.

o To further strengthen this point, supplementation of milk with purines or folate should be tested to determine whether this rescues the observed phenotypes.

2. Use of NCTC8325-4:

o The manuscript utilizes the human isolate NCTC8325-4 but does not provide sufficient justification for its selection, particularly in the context of milk growth. This strain is known to carry genetic mutations that affect transcriptional regulation, including SigB deficiency, as noted in https://pmc.ncbi.nlm.nih.gov/articles/PMC3786944/.

o SigB-deficient strains are associated with chronic udder infections, as described in https://www.nature.com/articles/s41598-019-49981-6 and https://www.frontiersin.org/journals/microbiology/articles/10.3389/fmicb.2019.02493/full. If the authors intentionally used NCTC8325-4 to model mastitis isolates, this should be explicitly discussed.

o The authors should address whether SigB deficiency is relevant to the phenotypes observed. The SH1000 strain, which represents a SigB-repaired variant of NCTC8325-4, could serve as a control. I would expect that SigB deficient strains show improved fitness in milk than WT strains. The fact that SigB-knowckdowns strains were not detected in the CRISPRi-seq analysis might further suggest that the used WT strain was already SigB deficient.

o Validation of the key findings using additional bovine and human S. aureus isolates is strongly recommended to ensure the results are generalizable.

3. Growth Defects of pur Kockdowns:

o Figure 3AB does not convincingly demonstrate growth defects for purA and purE. The differences between these strains and the wild type appear minor, and only purB shows a clear reduction in growth. Additionally, the purB colonies exhibit heterogeneity, raising questions about the method used to assess fitness.

o If colony-forming units (CFU) were counted from the dilution spot assay to generate the data for Figure 3B, this technique may lack precision, especially when differences between strains are relatively small (e.g., 2-fold rather than 10- or 100-fold). This likely explains the large standard deviations observed in Figure 2B.

o The authors should consider presenting raw CFU counts for different conditions to better assess variability and reliability of the experiment. The use of "relative fitness" in Fig. 3B is confusing, as this term is typically applied to competition assays involving co-culture of strains.

o For clarity, bar graphs should group strains based on growth conditions (e.g., all strains grown in milk paired together, and all strains grown in RPMI paired). This approach should be applied to Figures 4 and 5 as well.

Reviewer #2: Not applicable

Reviewer #3: An appropriate bovine-adapted strain should be examined to determine if the same genes/pathways identified in the human-derived lab strain 8325-4 are required for growth in milk. eg using mutants/knockdowns of selected genes.

Test the therapeutic efficacy of the folate inhibitor antimicrobials in milk eg trimethoprim-sulfamethoxazole with relevant bovine clinical isolates representing major lineages.

The limitations of the study with regard to the strain selection and its potential to provide insights into growth in milk and adaptation to the dairy environment should be clearly stated in the Methods and in the Discussion.

**Part III – Minor Issues: Editorial and Data Presentation Modifications**

Reviewer #1: 1. Supplementary Data Files:

o Supporting Tables S1 and S2 are not provided in the supplementary materials.

2. Text Corrections:

o Line 42: "remain".

o Line 64: "Colonization in staphylococci" – the meaning of this phrase is unclear and needs rephrasing.

o Line 72: "acid".

o Line 169: The reference to "Table 1" seems out of place, as the analysis forming the basis of this table has not yet been described.

3. Experimental Replicates:

o Several figures including Figure 1C, 3B 4A and 6, indicate that only two biological replicates were included. Three biological replicates are standard for publication and should be provided.

Reviewer #2: 1. Include more description and explanation of the CRIPR technique in the Introduction to make it easier for the reader to understand.

2. Expand on the osmoprotective effect of milk possibly being responsible for the diminished importance of teichoic acid biosynthesis in the Discussion.

3. Go down the rabbit hole of trying to compare the chemical compositions of milk and BHI (no fun I know).

Reviewer #3: The assertion of the authors that the study 'contributes to the understanding of bacterial adaptation in the dairy

environment' is not supported as the appropriate ecologically adapted strains are not considered.

PLOS authors have the option to publish the peer review history of their article (what does this mean? ). If published, this will include your full peer review and any attached files.

**Do you want your identity to be public for this peer review?** For information about this choice, including consent withdrawal, please see our Privacy Policy .

Reviewer #1: No

Reviewer #2: No

Reviewer #3: No

**Figure resubmission:**
---

## [Decision Letter · Decision Letter 1]

27 Mar 2025

Dear Dr. Kjos,

We are pleased to inform you that your manuscript 'Genome-wide analysis of fitness determinants of Staphylococcus aureus during growth in milk' has been provisionally accepted for publication in PLOS Pathogens.

Before your manuscript can be formally accepted we ask you to add important information to the abstract as suggested by reviewer #3 and you will need to complete some formatting changes, which you will receive in a follow up email. A member of our team will be in touch with a set of requests.

Best regards,

Andreas Peschel, Ph.D.

Academic Editor

PLOS Pathogens

Michael Otto

Section Editor

PLOS Pathogens

Sumita Bhaduri-McIntosh

Editor-in-Chief

PLOS Pathogens

orcid.org/0000-0003-2946-9497

Michael Malim

Editor-in-Chief

PLOS Pathogens

orcid.org/0000-0002-7699-2064

Reviewer Comments (if any, and for reference):

Reviewer's Responses to Questions

**Part I - Summary**

Reviewer #1: The authors provide a solidly revised manuscript. Additional datasets addressing my concerns are provided and appropriately incorporated into the manuscript.

Congratulations to a nice manuscript.

Minor:

Ln: 205 -L208: I suggest to add that this experiment suggest purines to represent limited nutrients within milk.

Reviewer #2: (No Response)

Reviewer #3: The authors have partly addressed the comments by testing the folate inhibitors in bovine mastitis strains which indicates this approach may have utility for treating mastitis. However, the authors did not make mutants in relevant bovine strains to test the relevance of their findings in appropriate strains. This remains a major limitation of the study. Currently there is no mention in the Title or abstract of the strain that is employed- ie the human-derived laboratory strain 8325-4- this should be addressed clearly in the abstract.

**Part II – Major Issues: Key Experiments Required for Acceptance**

Reviewer #1: (No Response)

Reviewer #2: (No Response)

Reviewer #3: (No Response)

**Part III – Minor Issues: Editorial and Data Presentation Modifications**

Reviewer #1: (No Response)

Reviewer #2: (No Response)

Reviewer #3: (No Response)

PLOS authors have the option to publish the peer review history of their article (what does this mean? ). If published, this will include your full peer review and any attached files.

**Do you want your identity to be public for this peer review?** For information about this choice, including consent withdrawal, please see our Privacy Policy .

Reviewer #1: **Yes: ** Simon Heilbronner

Reviewer #2: No

Reviewer #3: No

---

## [Editor Report · Acceptance letter]

Dear Dr. Kjos,

We are delighted to inform you that your manuscript, "Genome-wide analysis of fitness determinants of Staphylococcus aureus during growth in milk," has been formally accepted for publication in PLOS Pathogens.

Best regards,

Sumita Bhaduri-McIntosh

Editor-in-Chief

PLOS Pathogens

orcid.org/0000-0003-2946-9497

Michael Malim

Editor-in-Chief

PLOS Pathogens

orcid.org/0000-0002-7699-2064